# Leucoderma in Buffaloes (*Bubalus bubalis*) in the Amazon Biome

**DOI:** 10.3390/ani13101665

**Published:** 2023-05-17

**Authors:** José Diomedes Barbosa, Bruno Igor de Oliveira Possidonio, Janayna Barroso dos Santos, Hanna Gabriela da Silva Oliveira, Ananda Iara de Jesus Sousa, Camila Cordeiro Barbosa, Edsel Alves Beuttemmuller, Natália da Silva e Silva Silveira, Marilene Farias Brito, Felipe Masiero Salvarani

**Affiliations:** 1Instituto de Medicina Veterinária, Universidade Federal do Pará, Castanhal 68740-970, PA, Brazil; diomedes@ufpa.br (J.D.B.); igorpossidonio@outlook.com (B.I.d.O.P.); barrosojanavet@gmail.com (J.B.d.S.); hnnagabriela@gmail.com (H.G.d.S.O.); anandaiara@hotmail.com (A.I.d.J.S.); nataliasilvasilveira1@gmail.com (N.d.S.e.S.S.); 2Departamento de Epidemiologia e Saúde Pública (DESP), Instituto de Veterinária (IV), Universidade Federal Rural do Rio de Janeiro (UFRRJ), Seropédica 23890-000, RJ, Brazil

**Keywords:** acromotrichia, skin depigmentation, leather, copper, melanin, albinism, vitiligo

## Abstract

**Simple Summary:**

Bubalinoculture is a growing industry worldwide, and the commercialization of buffalo leather is rapidly expanding due to its thickness, weight, and flexibility compared to bovine leather. However, dermatological diseases in buffalo, such as leucoderma, are becoming increasingly common, particularly in the Amazon biome, resulting in significant economic losses due to the inability to utilize the leather. This study is the first to investigate and describe the epidemiological and clinicopathological aspects of leucoderma in buffalo in the Amazon biome, as well as to propose solutions to this problem in the region. The samples showed that there is no predisposition by breed, sex, or age for the occurrence of leucoderma. Regression of the clinical signs of acromotrichia and skin depigmentation occurred after a correct mineral supplementation of the animals, which is crucial for animal production systems. Mineral deficiency, particularly copper, is a significant predisposing factor for the occurrence of leucoderma in buffaloes in the Amazon biome. Therefore, the mineralization of buffalo herds is a continuous prophylactic practice that should be adopted by farmers afflicted with this problem.

**Abstract:**

Leucoderma is a condition that affects the skin and hair of animals, causing depigmentation and acromotrichia. In buffaloes, this condition results in significant economic losses for the production chain due to its impact on the leather trade. This study aimed to investigate the epidemiological and clinicopathological aspects of leucoderma in buffaloes in the Amazon biome and describe the prophylactic treatment to control the disease. The study included 40 buffaloes, 16 males and 24 females, aged between 1 and 10 years, and were of the Murrah, Jafarabadi, Mediterranean, and Murrah × Mediterranean crossbreed breeds. The animals were raised without mineral supplementation. The clinical signs observed in the animals included acromotrichia and depigmentation, with varying degrees and distribution of skin lesions. Histological examination of the epidermis showed interrupted melanin production, mild dermal fibrosis, mild perivascular mononuclear inflammatory infiltrate, and pigmentary incontinence. None of the animals had the genotype for albinism. After 120 days of mineral supplementation based on the use of copper sulfate, the clinical signs of leucoderma regressed. There was no predisposition by breed, sex, or age for the occurrence of the disease. The regression of skin lesions after proper mineral supplementation suggests that copper deficiency may be considered an important factor for the occurrence of leucoderma in buffaloes in the Amazon biome.

## 1. Introduction

As of 2022, the global buffalo population was estimated at 208 million heads [1]. In Brazil, the buffalo herd has already surpassed 2.5 million animals, with the largest number of buffaloes located in the Amazon biome. The State of Pará stands out, with over 619 thousand heads and an estimated gross income of about 200 million dollars for the production chain in the region [2]. Despite the growth of buffalo breeding in Brazil, it is not yet among the most important and nationally prominent livestock alternatives. This is due to farmers’ lack of knowledge regarding zootechnical and sanitary issues, as well as the anatomical and functional characteristics of buffaloes [3]. Buffaloes are known for their multiple aptitudes and are used for traction and the production of meat, milk, and leather. When compared to bovine-origin products, they have been considered equivalent or superior in terms of quality. For example, buffalo leather is widely used in the production of fine leather due to its greater flexibility, durability, and resistance [4,5]. Additionally, buffaloes have specific cutaneous characteristics, such as a high concentration of melanin, a low number of sweat glands, and a reduced concentration of hair. These specific characteristics provide a differential for buffalo leather, which significantly increases the importance of buffalo breeding, especially in humid tropics and developing countries [6].

However, buffalo skin, like that of other ruminants, is susceptible to various pathologies of immune-mediated, congenital, hereditary, hormonal, metabolic, nutritional, microbiological, chemical, physical, parasitic, and allergic nature. These pathologies result in significant economic losses for the production sector due to the costs of treatment, unpleasant external appearance, difficulty in commercializing the animals, and, most importantly, losses in the use of leather [7,8]. In this sense, leukoderma, a multifactorial disease that affects the tegumentary system of buffaloes, results from the loss of normal skin and hair pigmentation due to melanocyte alterations. It should be considered a disease of importance for the buffalo production system [3,9]. Recently, skin depigmentation and acromotrichia have been observed in buffaloes, especially in the Brazilian Amazon biome. However, no scientific study has been conducted to determine the possible cause of these tegumentary clinical signs or to differentiate whether these cases involve albinism, vitiligo, or leukoderma. Therefore, this study aimed to investigate the epidemiological and clinical-pathological aspects of the possible cases of leukoderma in buffaloes, as well as to describe the prophylaxis used to potentially control the disease.

## 2. Materials and Methods

Clinical care was provided to 40 buffaloes suffering from skin and hair depigmentation (acromotrichia) in five municipalities within the Brazilian Amazon biome. The animals belonged to four farms located in the municipalities of Mojú, Nova Timboteua, Castanhal, and Porto de Moz, in the State of Pará, and one farm was located in the municipality of São Matheus do Maranhão, in the State of Maranhão. During the veterinary doctor’s visit to the properties, epidemiological data, such as age, breed, sex, breeding system, feeding, use of mineral supplementation, vaccination schedule, and animal clinical history, were collected. Clinical examinations were performed according to Dirksen et al. [10]. The biopsies were taken from an affected skin area of all animals for histopathological study. The material was fixed in 10% buffered formalin and sent to the Pathological Anatomy Sector of the Federal Rural University of Rio de Janeiro (UFRRJ). The collected tissues were processed using the standard methods for histological examination, embedded and included in paraffin, cut at 5 μ, and stained with hematoxylin and eosin, and toluidine blue. Part of the biopsy material was refrigerated and used for a differential molecular diagnosis of albinism [11]. As a treatment attempt, the animals were administered mineral supplementation ad libitum, primarily containing copper sulfate and other minerals in the mineral mixture in the trough via the oral route. Only one animal in the study underwent a pharmacological therapeutic protocol. This research was authorized by the animal experimentation ethics committee (CEUA) of the Federal University of Pará (UFPA) under protocol number 1480260522 (ID 001994).

## 3. Results

Out of the 40 animals that were treated, 16 were male and 24 were female. They belonged to the Murrah, Jafarabadi, Mediterranean, and crossbreed of Murrah with Mediterranean breeds and were between the ages of 1 and 10 years. The farms where the study was conducted used an extensive breeding system without mineral supplementation. During the rainy season, the animals were kept in Urochloa (Brachiaria) sp. pastures and native vegetation pastures. In the dry season, they were kept in floodplain areas where only low-nutritional and mineral native vegetation was available. All four farms had a history of buffaloes with recurrent clinical symptoms, such as whitish hair (acromotrichia) and depigmentation of the skin with a white-brownish/white-pinkish appearance that turned milky-white with an aged appearance over time. The lesions were generally well-defined, with irregular contours, asymmetrical, unilateral or bilateral, located in various regions of the body, and even coalescent (Figure 1). In some animals, acromotrichia and depigmentation regressed within weeks, while in others, they persisted for months. In one animal, the loss of pigmentation was almost total, making it hypothetically “albino.”

During the histological examination of the epidermis, it was observed that there was a disruption in the presence of melanin, as well as mild dermal fibrosis, perivasculitis, mononuclear perianexitis, and pigment incontinence (Figure 2). Molecular genetic research revealed that none of the buffaloes studied had the genotype for albinism, indicating that none of the animals had a mutation in the tyrosinase gene protein (TYR). 

As a treatment attempt, the animals were administered mineral supplementation ad libitum, primarily containing copper sulfate at a ratio of 406 mg/kg of copper in the mineral mixture in the trough. After a minimum of 90 days of mineral supplementation, the clinical signs of leucoderma regressed to the point where areas of acromotriquia and depigmentation of the skin were no longer noticeable. However, in one animal, the clinical signs did not completely disappear after mineral supplementation. Therefore, a pharmacological therapeutic protocol was implemented, which involved the use of the corticosteroid dexamethasone, given at a dosage of 5 mg/kg intramuscularly for 30 days, in conjunction with copper supplementation. This resulted in the complete cure of the disease (Figure 3).

## 4. Discussion

The results of this study indicate that leucoderma, which was first diagnosed in the Amazonian biome, occurred in buffaloes of different breeds, crossbreeds, sexes, and ages. This suggests that there is no breed, sex, or age predisposition for the onset of the disease, which is consistent with the findings from previous studies conducted in India by Singh and Randhawa [12] and Gapat et al. [13]. However, Randhawa et al. [14] reported that the disease is more likely to occur in buffalo heifers due to their musculoskeletal system development, which increases their nutrient requirements, including macro- and micronutrients such as copper. Copper deficiency can lead to the occurrence of leucoderma, as this microelement is involved in multiple enzymes, including tyrosinase, and various physiological functions. It is also possible that other minerals, such as molybdenum [15], may interfere with the development of the disease.

The exact cause of the disease is not fully understood, but it is believed that the appearance of skin and hair depigmentation is directly linked to a malfunction in the metabolism of the enzyme tyrosinase, which is dependent on copper ions. This is because a decrease in copper conversion, present in the active site of tyrosinase, the key enzyme in skin pigmentation, impairs the synthesis of melanin. Therefore, depending on the dietary sources of this micronutrient or the presence of antagonistic elements in the animal’s diet, the absorption of copper, which occurs mainly in its bivalent form in the small intestine, may vary and, consequently, lead to the development of leukoderma [15,16,17]. Higher levels of molybdenum result in thio-molybdenum complexes, which render copper metabolically unavailable to the animal system, preventing its use in tyrosinase synthesis. As a result, diets rich in molybdenum can induce hypocuprosis and possibly the appearance of animals with leukoderma [15].

The Brazilian states of Pará and Maranhão, situated in the Amazon biome, were the locations of the present study. These regions are known for their deficiencies in phosphorus, copper, and cobalt in both cattle and buffalo [18,19,20]. Pinheiro et al. [21] conducted a study on Marajó Island, located in the State of Pará, and found low levels of copper, zinc, and cobalt in the liver tissue of buffalo of various ages and sexes. The authors also noted that all the studied animals had copper and zinc concentrations below critical levels, indicating a severe deficiency of these minerals. For instance, the copper levels in liver tissue ranged from 5.57 to 7.60 ppm, which is significantly lower than the adequate reference levels of 100 ppm [21,22]. These findings suggest that copper deficiency may play a role in cases of leucoderma in buffalo, as the 40 animals studied were kept in an extensive breeding system without mineral supplementation, which could be directly related to a possible copper deficiency. However, the measurement of minerals in animals, pastures, and soil was not the focus of the present study, and this perspective is crucial for future research on the subject.

Leucoderma is characterized by a varied distribution of lesions that can be localized, extensive, or multifocal [12,13,14]. In this study, we observed a variation in the location of skin and hair lesions, with some animals showing clinical signs restricted to a specific region of the body while others had multifocal or locally extensive lesions. Among the buffaloes studied, leucoderma was classified as localized in 24 animals, multifocal in 16, and only 1 animal presented almost complete body leucoderma, which was characterized as “pseudo-albino.” In some cases, focal lesions were extensive and varied in size. The mechanisms that can lead to hypopigmentation of the skin and hair include a decrease in melanin production, a reduction in the dispersion of melanin granules, and an increase in melanin loss [7,8,9]. In the animals of this study, we observed melanin loss, as the animals had normal pigmentation and later showed depigmented regions of the body in the herd. This was confirmed in histopathology, where we observed the interruption of melanin presence, as well as dermal fibrosis, mononuclear perivasculitis and perianexitis, and pigment incontinence. This inflammatory process of the affected skin may be directly related to a reaction of the skin unprotected by melanin and exposed to an environment of intense sun exposure, as occurs in the Amazon biome.

Accurate differential diagnosis of albinism is crucial, as it is a congenital disorder characterized by either a generalized or partial depigmentation of the skin, which is also a common clinical symptom of leucoderma. However, albino animals are born with the condition and retain it throughout their lives, unlike leucoderma, where depigmentation may disappear after some time [11,23] or after treatment. Despite their similarities, albinism and leucoderma have different origins. Albinism is an autosomal recessive genetic anomaly that reduces or eliminates melanin production due to a point mutation in the decoding of the TYR gene [23]. None of the animals in the study were diagnosed with a TYR gene mutation, which supports the diagnosis of leucoderma. Leucoderma is often confused with vitiligo, an autoimmune disease that promotes depigmentation of the skin and hair by destroying melanocytes [24]. Some authors suggest that vitiligo’s pathogenesis is related to a non-inflammatory response by T lymphocytes against melanocytes [25,26]. It should also be noted that at least three different allele genes are involved in the pathogenesis of vitiligo, making it a polygenic disorder [27,28]. Therefore, leucoderma and vitiligo have very different physiopathologies. The depigmentation of hair and skin in leucoderma is directly associated with metabolic problems with tyrosinase and copper deficiencies [13,14,24,25,28].

Prophylaxis for buffaloes displaying symptoms of leucoderma involved introducing a mineral supplement into their diet containing 250 g of copper sulfate per 100 kg of the mineral mix. With an estimated daily consumption of 60 g per animal, individual ingestion of approximately 100 mg of copper was achieved, resulting in a clinical cure for the animals, with leucoderma disappearing between 90 to 120 days. The mineral supplementation in this study was a primary corrective and prophylactic action, adopted based on literature data indicating a copper deficiency in the soil and animals in the Amazon biome [18,19]. This treatment strategy differed from that adopted by Gapat et al. [13] in India, who only used the oral administration of copper sulfate at a concentration of 300 mg/100 kg of body weight (BW), with a clinical cure for the 18 animals occurring, varying from 55 to 210 days. Another treatment method was recommended by Varum et al. [29], who administered a subcutaneous injection of 10 mL containing zinc (60 mg/mL), manganese (10 mg/mL), selenium (5 mg/mL), and copper (15 mg/mL) every 7 days for 60 days in each animal that had clinical signs compatible with leucoderma, resulting in a clinical cure of 87.5% (7/8) of the animals at the end of the 60-day treatment period. Although both individual oral treatment and subcutaneous injection administration were effective, they are considered impractical from a management perspective on farms. Therefore, mineral supplementation in the feed trough is the best strategy for the prophylaxis and control of leucoderma and other mineral deficiencies. Within 90 days, the buffaloes in this study showed improved clinical signs, and at 120 days, the mineral supplementation had effectively worked in 39 of the 40 animals, which no longer showed achromotrichia and whitish skin. Although mineral treatment was effective in almost all animals, one animal (2.5%, 1/40) still showed depigmentation of the skin. For persistent symptoms in animals, the literature [30] recommends exclusive intramuscular corticosteroid therapy using dexamethasone at a dosage of 5 mg/kg/BW for 30 days. In this study, the use of 5 mg/kg/BW of dexamethasone for 30 days did not produce the desired effect. However, after changing the treatment regimen to a double dose of corticosteroid and mineral supplementation for 30 days, an improvement was observed. There were no longer clinical symptoms of leucoderma diagnosed in this animal. 

## 5. Conclusions

This study is the first to explore leucoderma in buffaloes in the Amazon biome. The disease was found to affect buffaloes of both sexes as well as all ages and breeds. Additionally, it was discovered that a copper deficiency may have played a significant role in the animal depigmentation process, as the use of mineral supplementation in the herds resulted in a positive response and a complete clinical recovery without relapse. Therefore, continuous mineralization of buffalo herds should be adopted as a prophylactic management practice by farmers who face this issue, mainly in the Amazon region.

## Figures and Tables

**Figure 1 animals-13-01665-f001:**
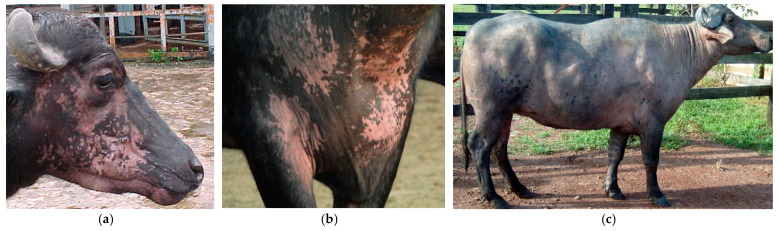
Leucoderma in adult Murrah buffaloes found in the Brazilian Amazon biome. The animals exhibit varying degrees of depigmentation in their skin and hair throughout their entire body: (**a**) on the animal’s face, (**b**) in the neck region with asymmetric and disordered patterns of smaller and larger proportions, and (**c**) almost complete depigmentation of the animal.

**Figure 2 animals-13-01665-f002:**
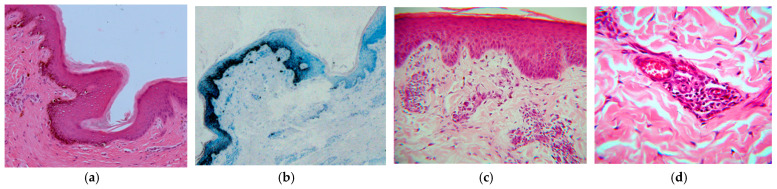
Leucoderma in buffaloes within the Brazilian Amazon biome. (**a**) Area where melanin is absent in the epidermis and mild orthokeratotic hyperkeratosis (Obj. 16, H.E.); (**b**) epidermis with a disruption in melanin production (Obj. 10, toluidine blue); (**c**) epidermis with no melanin pigment, mild fibrosis of the superficial dermis, and mononuclear perivascular inflammation (Obj. 25, H.E.); (**d**) detailed view of the mononuclear perivascular inflammation (Obj. 40, H.E.).

**Figure 3 animals-13-01665-f003:**
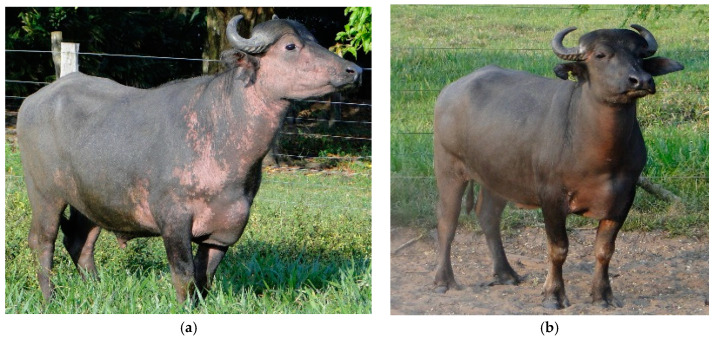
Leucoderma in crossbred Murrah and Mediterranean buffalo in the Amazon biome: (**a**) treated solely with mineral supplementation; (**b**) after treatment with a combination of corticosteroid and copper supplementation.

## Data Availability

Not applicable.

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
