# Peer review of "Leucoderma in Buffaloes (Bubalus bubalis) in the Amazon Biome"

_animals, 2023, doi:10.3390/ani13101665_

Round 1

Reviewer 1 Report

The authors presented a case study with 40 buffaloes showing leucoderma. The report is interesting, very well written, and adequate to publishing in ANIMALS. I do not have major concerns, only a few minor edits. Congratulations to the authors.

Minor comments

L136: Can you clarify what you mean by supplementation? Was this copper supplemented in the diet? Was the copper injected? Please describe in detail how and when the copper was provided. If fed, what was the carrier? I could see more details in the discussion, but I recommend adding them in M&M as well.

L159: Change "nutrient intake needs" to "nutrient requirements"

L182: "copper levels in liver tissue ranged..."

L348: Bold "1998"

Author Response

The authors are very grateful for the comments, praise, criticisms and suggestions of reviewer 1 and hereby inform the following changes:

1). The information was entered into the M&M. And we emphasize that copper supplementation was done orally, but not forced or introduced by a veterinarian into the oral cavity. The animals had the mineral mixture available at will in the trough, and licked according to their nutritional needs. In previous studies conducted by our research group in the Amazon biome, the concentration of copper sulfate at a ratio of 406 mg/kg of copper in the mineral mixture is sufficient to ensure that each animal is able to ingest the appropriate proportion of copper per kg, which, as demonstrated the experiment was sufficient to correct the deficiency of the mineral and consequently the occurrence of leucoderma. 

2). Change made.

3). Inserted the words "in liver tissue".

4). Change made.

Reviewer 2 Report

Introduction. Please add some notes regarding the occurrence of the pathological condition in other animal species. Also, please add some text regarding the pathogenetic background of the disorder.

Clinical work. Please add microphotographs of normal skin of buffaloes and please comment on the differences between the two types of skin.

Regarding the administration of copper for therapeutic purposes, as there are results of blood biochemistry, please discuss your reasoning for this therapeutic approach.

Discussion. The discussion is on the longer side, in relation to the remaining manuscript, hence, please decrease in length.

An interesting case-report, which should be corrected before possible acceptance.

Author Response

The authors are very grateful for the comments, praise, criticisms and suggestions of reviewer 2 and hereby inform the following changes:

Introduction: the work was initially submitted as an original article, but according to the journal's editors it was reformulated to fit in as a short communication. So the introduction was quite direct and concise. Humbly in the opinion of the authors, inserting data from other species in the introduction would not add useful scientific information to be used in the discussion, as it would not be possible to compare the disease in different species, even if the etiopathogenesis were the same. The focus of the work was really to inform the scientific community about the occurrence of leukoderma in buffaloes in the Amazon Biome, as the first report in the world.

We chose to make a broader approach to skin diseases in buffaloes in the introduction and better discuss the pathogenetic background of the disorder in the discussion section, where we can better explain the differences between leucoderma, albinism and vitiligo. Precisely for this reason, the discussion of the article, in our view the most important part of the article, was longer than the other parts that are part of the structure of a scientific text (Introduction, Material and Methods, Results, Discussion and Conclusion). We are really sorry if we were unable to fully respond to your suggestions, but I hope you understand our position because for us authors, the way the article is presented and structured makes the article better from a scientific point of view, as pointed out by other reviewers and even editors of Animals magazine, who support our view.

Clinical work: we inform you that blood was not collected or other tissues such as spleen, liver and bones to measure different minerals, including copper. Therefore, unfortunately we do not have results for of blood biochemistry. Our reasoning for this therapeutic approach based on mineral supplementation was based on the molecular differential diagnosis for albisnism, on the epidemiological findings (animals of different races, ages and sexes), clinical signs consistent with leukoderma and mainly because they are animals raised in an extensive system, with based on forages of low nutritional and mineral value, in addition to not using any of the properties of continuous mineral supplementation by trough. This led us to infer that it could be a deficiency disease, based on copper deficiency, and therefore the initial therapeutic approach was to use as a treatment attempt, the animals were administered mineral supplementation primarily containing copper sulfate and other minerals in the mineral mixture in the trough, oral route, ad libitum.

Regarding the insertion of microphotographs of normal and leucordma skin of buffaloes and their differences, it will not be possible, as our work does not have this objective and focus, since there are already articles and specific books on dermatology in the scientific literature, which already show these types of of comparison and differences. What we did in figure 2 was to show what (a) area where melanin is absent in the epidermis and mild orthokeratotic hyperkeratosis (Obj. 16, H.E.); (b) epidermis with a disruption in melanin production (Obj. 10, toluidine blue); (c) epidermis with no melanin pigment, mild fibrosis of the superficial dermis, and mononuclear perivascular inflammation (Obj. 25, H.E.); (d) detailed view of the perivascular mononuclear inflammation. Histopathological findings distinct from intact skin, as, I reiterate, can be found in dermatology books and articles.

Below are some references:

  • Montage, The Structure and Function of Skin, Academic press, New York, NY, USA, 2nd edition, 1960.
  • Singh A. Skin Pigmentation in Buffalo Calves. Can Vet J. 1962 Nov;3(11):343-6. PMID: 17421544; PMCID: PMC1585998.
  • S. E. Hafez, A. L. Badreldin, and M. M. Shafei, “Skin structure of Egyptian buffaloes and cattle with particular reference to sweat glands,” The Journal of Agricultural Science, vol. 46, no. 1, pp. 19–30, 1955.
  • R. Saravanakumar and M. Thiagarajan, “Comparison of sweat glands, skin characters and heat tolerance coefficients amongst Murrah, Surti and non-descript Buffaloes,” Indian Journal of Animal Sciences, vol. 62, pp. 625–28, 1992.
  • D. Dellmann, Textbook of Veterinary Histology, Lea and Febiger, Philadelphia, Pa, USA, 4th edition, 1993.
  • https://doi.org/10.33899/ijvs.2009.5741
  • Elhaig MM, Selim A, Mahmoud M. Lumpy skin disease in cattle: Frequency of occurrence in a dairy farm and a preliminary assessment of its possible impact on Egyptian buffaloes. Onderstepoort J Vet Res. 2017 Mar 28;84(1):e1-e6. doi: 10.4102/ojvr.v84i1.1393. PMID: 28397517; PMCID: PMC6238723.
  • Debajit Debbarma, Varinder Uppal, Neelam Bansal, Anuradha Gupta, "Histomorphometrical Study on Regional Variation in Distribution of Sweat Glands in Buffalo Skin", Dermatology Research and Practice, vol. 2018, Article ID 5345390, 7 pages, 2018. https://doi.org/10.1155/2018/5
  • Liang Z, Yao K, Wang S, Yin J, Ma X, Yin X, Wang X, Sun Y. Understanding the research advances on lumpy skin disease: A comprehensive literature review of experimental evidence. Front Microbiol. 2022 Nov 28;13:1065894. doi: 10.3389/fmicb.2022.1065894. PMID: 36519172; PMCID: PMC9742232.

Discussion: the discussion section of an article is, in the humble point of view of these authors, the most important part of the article. It is in the discussion that we are able to better explain all the information addressed in the article, such as comparing it with other studies, inferring opinions and detailing the scientific opinions and criticisms of the group. In our view, there is no discrepancy that the discussion is too long, since all paragraphs are important and described in detail so that the scientific community and even academics of veterinary medicine and veterinary medical professionals understand our article, our therapeutic choices, our epidemiological and clinicopathological findings. Unfortunately, we cannot respond to your suggestion, as, like other reviewers and editors, the discussion is adequate to the reality of this short communication.

We would like to inform you that the article was translated and revised by the company Editage, whose certificate is attached.

Reviewer 3 Report

The topic of the paper is interesting, but I would like to get some more data from the authors on animal recruitment, how many animals had leukoderma? The authors say this is the first study to investigate and describe the epidemiological and clinicopathological aspects of leukoderma in buffalo in the Amazon biome. Can the authors detail these aspects in more details?

Author Response

The authors are very grateful for the comments, praise, criticisms and suggestions of reviewer 3 and hereby inform the following changes:

1). All 40 animals had clinical symptoms compatible with leukoderma. And with the molecular differential diagnosis for albinism, we can buy that all 40 animals had leucoderma and so they were all entered into the study. The epidemiological and clinicopathological aspects are described throughout the text, but we reiterate that it is a disease that affects animals of any breed, sex and age, located in different regions of the Amazonian biome, reared in an extensive system, with forages of low nutritional quality, without continuous mineral supplementation (very common in buffalo farming in the region), herds that show whitish hair (acromotrichia) and depigmentation of the skin with a white-brownish/white-pinkish appearance that turned into milky white with an aged ap-pearance over time. The lesions were generally well-defined, with irregular contours, asymmetrical, unilateral or bilateral, located in various regions of the body, and even coalescent (Figure 1 of the article). During the histological examination of the epidermis, it was observed that there was a disruption in the presence of melanin, mild dermal fibrosis, perivasculitis, mononuclear perianexitis and pigment incontinence (Figure 2 of the article). In the discussion more details are mentioned and compared with other studies, mainly Indian ones, which also describe well the occurrence of the disease, however the big difference is that in the current study we describe and present the histological lesions and we do the treatment via mineral supplementation to trough and not by intramuscular injection. In this work we show the importance of preventive management not only of leucoderma, but of other deficiency diseases through the adoption of mineral supplementation as a common practice within buffalo breeding farms. We hope we were able to answer and clarify your questions, thank you again for your comments.

Round 2

Reviewer 2 Report

Before acceptance, the authors should add a new paragraph in the discussion, regarding the consequences of the problem at population level in the area of the study.